# Extraction of ‘Gannanzao’ Orange Peel Essential Oil by Response Surface Methodology and its Effect on Cancer Cell Proliferation and Migration

**DOI:** 10.3390/molecules24030499

**Published:** 2019-01-30

**Authors:** Ke Liu, Weihui Deng, Wei Hu, Shan Cao, Balian Zhong, Jiong Chun

**Affiliations:** National Navel Orange Engineering Research Center, College of Life Sciences, Gannan Normal University, Ganzhou 34100, China; liuke121602026@126.com (K.L.); dwh110by@163.com (W.D.); weihu2919@163.com (W.H.); scoral29116@163.com (S.C.); bal.zh@163.com (B.Z.)

**Keywords:** ‘Gannanzao’ orange, essential oil, hydrodistillation, response surface methodology, GC-MS, anticancer activity

## Abstract

The essential oil of ‘Gannanzao’ orange peel was extracted by hydrodistillation, and the extraction conditions were optimized by Box–Behnken response surface methodology. The components of essential oil were analyzed by GC-MS. Thirty-nine different components were detected, accounting for 99.59% of the total oil. Limonene (88.07%) was the prominent component. The optimal extraction conditions were as follows: liquid material ratio of 8.4:1 (mL/g), sodium chloride concentration of 5.3%, and distillation time of 3.5 h. The Cell Counting Kit-8 assay showed that ‘Gannanzao’ orange peel essential oil had good dose-dependent inhibition effect on the proliferation of HepG2 hepatoma cells and HCT116 colorectal cancer cells. When the concentration of the essential oil was 0.6 μL/mL or higher, the viability rate of both cancer cells became lower than 13.0%. The transwell assay indicated the essential oil can inhibit migration of both cancer cells at the concentration of 0.3 μL/mL.

## 1. Introduction

Cancer is a serious disease that threatens human health with a high mortality rate. It is characterized by uncontrolled growth and spread of abnormal and proliferating cells that have undergone a plethora of changes in multiple genes [1]. Chemoprevention is the hot spot of cancer control, which can impede or slow the development of cancer by pharmacological agents and reduce the incidence and mortality of cancer [2,3]. As some anticancer drugs have high toxicity and resistance rates and are ineffective in some types of tumors, the research and development of greater resources towards prevention are needed. Natural anticancer agents from plants have drawn much interest and played a significant role in cancer drug discovery due to their multitude effects on diverse molecular signaling pathways, with no or minimal toxicity in normal cells. Newman et al. reported [4] that, from the 1940s to 2014, 75% of the 175 small molecule anticancer drugs approved by the US Food and Drug Administration (FDA) are actually either natural products or their derivatives.

Essential oil (EO) is a kind of natural product found in the roots, stems, leaves, flowers, and fruit peels of aromatic plants. It is mainly composed of terpenes, alcohols, aldehydes, ketones, acids, and esters [5]. Citrus EO is a kind of widely used plant EO, which is mainly rich in fruit peel. Studies have reported that citrus EOs have a variety of biological activities, including antioxidant, bacteriostatic, anti-inflammatory, anticancer, and insecticidal activities [6,7,8,9,10]. Sahay [11] reviewed the pharmacological effects of EOs which have antibacterial and anticancer activities. EOs have little toxicity and few side effects, and are generally recognized as safe in food, and may become new natural resources for the chemoprevention of cancer [12,13].

Orange is one of the most important fruit tree crops in the world. Brazil and China are the world’s leading producers of orange, and the production in 2017/18 is about 16.03 and 7.30 million tons, respectively [14]. ‘Newhall’ navel orange (*Citrus sinensis*) is broadly cultivated in the Gannan region of Jiangxi province in China, with a total annual output of more than one million tons. ‘Gannanzao’ is a new variety selected from the bud mutation of ‘Newhall’ navel orange, which has the characteristics of early maturity (the harvest time is one month earlier than the latter), excellent flavor, and easy peeling [15,16]. A large amount of peel is produced during orange processing. It contains several functional natural products, such as EO, pectin, carotenoids, and hesperidin, etc., which are important raw materials in the chemical and pharmaceutical industries [17]. Exploring efficient methods to extract EO from orange peel and provide value-added products has become an important part of the citrus processing industry. In the present study, we optimized the hydrodistillation conditions for extracting EO from ‘Gannanzao’ orange peel by response surface design and identified its components. Our previous research [18] found that ‘Newhall’ navel orange EO has a significant inhibitory effect on human lung cancer cell line A549 and prostate cancer cell line 22RV-1. As well as we know, there are no literature reports on extraction and anticancer activities of ‘Gannanzao’ orange essential oil (GOEO). A preliminary study of its proliferation and migration effects on HepG2 hepatocellular carcinoma cells and HCT116 colon cancer cells was carried out. Our research results will provide information for efficient extraction of GOEO and potential utilization of it as a source of natural anticancer agent.

## 2. Results and Discussion

### 2.1. Response Surface Test Results and Analysis

Based on single effect experiments and literature reports, we designed three-factor, three-level experiments involving liquid material ratio (A), NaCl concentration (B), and distillation time (C). The experiments were carried out based on Box–Behnken design via Design-Expert 8.0.6 software (Stat-Ease Inc., Minneapolis, MN, USA), and the extraction yields of GOEO under different conditions are shown in Table 1.

Second-order polynomial equation was applied to express the yield of GOEO as a function of these coded variables:Y = 2.00 +0.065A + 0.088B + 0.11C + 0.037AB + 0.048AC + 0.022BC − 0.16A^2^ - 0.22B^2^ +0.00475C^2^(1)

The analysis of variance (ANOVA) is shown in Table 2. The model was highly significant (*p* < 0.001), indicating the equation was adequate for predicting the yield under any combination of values of the variables. A, B, C, A^2^, and B^2^ were highly significant model terms because *p*-values were less than 0.001. AB, AC, and BC were very significant model terms because *p*-values were less than 0.01. However, C^2^ was not significant because the *p*-value was greater than 0.05.

The lack-of-fit measured the failure of the model to represent data in the experimental domain at points which were not included in the regression. The *p*-value of 0.9269 implied it was not significant relative to the pure error, revealing that the quadratic model was statistically significant for the response.

The experimental data matched the model well with high correlation coefficients (R^2^ = 0.9952), indicating that only 0.48% of the total variations could not be explained by the model. The Pred R^2^ of 0.9857 was in reasonable agreement with the value of the adjusted determination coefficient Adj R^2^ of 0.9890. Adeq precision that represented the signal to noise ratio was greater than 4, indicating this model could be used to navigate the design space.

In this model, the *p*-values of liquid material ratio (A), NaCl concentration (B), and distillation time (C) were less than 0.001, indicating that the three factors had a highly significant effect on the yield of GOEO. It could be seen from the coefficients of the regression equation that the distillation time had the greatest influence on the response value of the yield, followed by NaCl concentration and the liquid material ratio (C > B > A). The interaction terms of factors *p*-values AB < 0.01, AC < 0.01, BC < 0.05, indicating that the interaction terms AB and AC had a very significant effect on the extraction yield, and the interaction term BC had a significant effect.

Figure 1 showed that the interaction term of each factor had a significant effect on the yield of EO. As shown in graph a, when distillation time C was kept constant at 3.0 h, as liquid material ratio and NaCl concentration increased the extraction yield also increased; at a certain point, the yield reached the highest value. However, when the liquid material ratio and NaCl concentration further increased, the yield began to decline. As shown in graph b, when NaCl concentration B was kept constant at 5.0%, with the increase of liquid material ratio, an upward trend of the yield was shown. The yield began to decline with further increase of liquid material ratio. However, the yield always increased with the increase of distillation time. As shown in graph c, when liquid material ratio A remained at 8:1, the yield was increased with the increase of distillation time. With the increase ofNaCl concentration, an upward trend of the yield was shown. The yield began to decrease with further increase of NaCl concentration. Using the response surface design, interaction of factors affecting the experiment could be considered simultaneously.

The model was optimized, and the predicted maximum value (2.15%) of the test variables was found under the following conditions—liquid material ratio of 8.38:1 (mL/g), NaCl concentration of 5.29%, and distillation time of 3.50 h. For the feasibility of the operation, liquid material ratio was adjusted to 8.4:1, and NaCl concentration to 5.3%. Experiments were performed under new conditions to validate the adequacy of optimized prediction. A value of 2.14% ± 0.03% (*n* = 3) demonstrated the validation of the RSM model. Ferhat et al. [19] used traditional hydrodistillation and microwave-accelerated distillation method to extract orange peel EO with yield of 0.39% and 0.42%, calculated based on the fruit weight or 1.95% and 2.10% based on peel weight, respectively. The different yields may be due to the different variety of oranges and extraction method.

### 2.2. Analysis of Essential Oil Components by GC-MS

The components of GOEO were analyzed by GC-MS. The total ion chromatogram (TIC) was shown in Figure 2. The relative content of each component was calculated by the peak area normalization method. The components were identified according to retention index and the NIST mass spectral library.

As shown in Table 3, thirty-nine components were detected by gas chromatography–mass spectrometry with a total content of 99.59%, which included eleven monoterpenes (limonene, β-myrcene, α-pinene, etc.), and twelve oxygenated monoterpenes (linalool, α-terpineol, geraniol, neral, etc.), accounting for 96.00% and 1.96%, respectively; seven sesquiterpenes (β-elemene, β-farnesene, valencene, etc.) and four oxygenated sesquiterpenes (β-sinensal, α-sinensal, nootkatone, etc.), accounting for 0.91% and 0.37%, respectively. In addition, five straight-chain aldehydes were identified, such as octanal, nonanal, decanal, etc., with a total relative content of 0.35%. Compared with ‘Newhall’ orange EO (NOEO) prepared by hydrodistillation, the total amount of monoterpenes in GOEO was 95.94%, higher than NOEO (89.87%). Limonene content of GOEO was about 6.4% higher than NOEO (81.62%). Percentage of oxygenated monoterpenes in GOEO (1.96%) was much lower than NOEO (6.94%). Linalool (0.26%) content in GOEO was much lower than NOEO (2.03%). Sesquiterpenes and oxygenated sesquiterpenes presented very low content in both EOs (0.81% and 0.17% for NOEO, 0.91% and 0.37% for GOEO, respectively). Compared with Valencia orange peel EO (VOEO) [19], limonene content of GOEO (88.07%) was about 10% higher than VOEO (78.5%), and β-myrcene content of GOEO (4.93%) was lower than VOEO (5.3%). The composition of these two orange EOs varied due to differences in variety, extraction method, and other factors.

### 2.3. Antiproliferative Activity of GOEO in HepG2 and HCT116 Cancer Cells

The effects of different concentrations of GOEO on the proliferation of HepG2 liver cancer and HCT116 colon cancer cells were tested by CCK-8 method. The results were shown in Figure 3. As the concentration of GOEO increased, the viability rate of both cells decreased. When the concentration of GOEO was higher than 0.3 μL/mL, the viability rate of HepG2 cells was significantly decreased; when the concentration of GOEO was 0.5 μL/mL or higher, the viability rate became lower than 13.2%. GOEO also had a good inhibitory effect on the growth of HCT116 colon cancer cells. When the concentration was 0.6 μL/mL or higher, the viability rate of HCT116 cells was lower than 13.0%. GraphPad Prism™ (Version 5.00) software (GraphPad Software, San Diego, CA, USA) was used to calculate IC_50_ values. IC_50_ value of HepG2 and HCT116 was 0.29 and 0.35 μL/mL, respectively. These results indicated that GOEO has a good inhibitory effect on the proliferation of HepG2 hepatoma cells and colon cancer HCT116 cells in vitro.

GOEO can inhibit the migration of HepG2 liver cancer and HCT116 colon cancer cells. Pre-experimental results showed that GOEO at a concentration higher than 0.3 μL/mL will kill most cancer cells. When the GOEO concentration was lower than 0.3 μL/mL, there was no obvious inhibition on the migration of cancer cells. Therefore, GOEO at concentration of 0.3 μL/mL was used to treat cancer cells. As shown in Figure 4, GOEO caused less migration of HepG2 hepatoma cells and HCT116 colon cancer cells compared with the control group, and the inhibitory effect was very significant (*p* < 0.01). The inhibition rate of HepG2 liver cancer cells was 55.23%, higher than that of HCT116 colon cancer cells (49.63%).

Studies have reported that plant EOs as well as their components such as limonene, citral, β-elemene can effectively inhibit the proliferation of various malignant cells such as melanoma, breast cancer, colon cancer [20,21,22]. Manassero et al [23] tested antiproliferative activity of EO from mandarin peel and its principal component limonene, and found that both EO and limonene showed a strong dose-dependent antiproliferative activity of lung adenocarcinoma A549 and hepatocarcinoma HepG2 cell lines. Liver cancer and colon cancer are common malignant tumors. The occurrence and development of tumors are not only related to cell proliferation, but also closely related to the migration and invasion ability of cancer cells. Therefore, anticancer drugs not only need to inhibit the proliferation of cancer cells, but also their metastasis. Our study preliminarily tested the inhibitory effect of GOEO on the proliferation and migration of HepG2 liver cancer cells and HCT116 colon cancer cells. The discussion about anticancer activity of some EO components has been made by Zengin et al [24]. The anticancer activity of GOEO and its components on cancer cells and their mode of action deserve further study.

## 3. Materials and Methods

### 3.1. Experimental Materials and Pretreatment

‘Gannanzao’ oranges were picked in October, 2017, from the germplasm resources of Gannan Normal University, Ganzhou City of Jiangxi Province, China (25°48′ N, 114°52′ E). Oranges were peeled by hand, separating the external part of the orange (flavedo), giving a yield of 20% (*w*/*w*) of orange peel with respect to the whole fruit. The fresh flavedo was crushed and used immediately as a sample in all extractions.

### 3.2. Extraction of Essential Oil by Hydrodistillation

Sample (100.0 g) was placed in a 2.0 L round bottom distillation flask, and a certain amount of sodium chloride (NaCl) and distilled water were added. The mixture was slowly heated to boiling, and distilled for a certain period of time. The distillate was collected, extracted three times with petroleum ether. The organic layers were combined, dried over anhydrous sodium sulfate, and then the petroleum ether was removed by a rotary evaporator. The extraction yield of essential oil (Y) was calculated as follows: Y (%) = (essential oil mass/flavedo mass) × 100(2)

### 3.3. Response Surface Design

The response surface methodology (RSM) has been successfully applied to optimize conditions in the extraction of essential oils [25,26]. Common process for optimization consisted of varying one parameter and keeping the others at a constant level. This single variable optimization neglected the interactive effects between variables thus the net effect of variables on the response was not exhibited. RSM has been designed to overcome this problem. It allowed simultaneous variation of several factors to find the optimal conditions for the desired response and reduced experimental trials needed to evaluate multiple parameters and their interactions [27].

There were many factors affecting the hydrodistillation extraction yield of EO [19], for example, liquid material ratio, distillation time etc. Yu et al. [28] has proofed the yield of EO can be enhanced by adding an appropriate amount of inorganic salts during the extraction process. Adding salt in the solution could change the composition of the original vapor-liquid equilibrium by association and other interactions in the liquid phase, which causes a ‘salt effect’ [29]. Based on single effect experiments and literatures, we designed three-factor, three-level experiments involving liquid material ratio (A), NaCl concentration (B), and distillation time (C). The range of independent variables and their levels are shown in Table 4. The test variables were transformed to the range of −1 to 1 for the evaluation of factors. The extraction conditions were optimized by Box-Behnken design via Design-Expert 8.0.6. 

### 3.4. GC-MS Analysis

‘Gannanzao’ orange essential oil (GOEO) components were analyzed by an Agilent 7890B gas chromatograph coupled with an Agilent mass spectrometer detector (Agilent Technologies, Santa, Clara, CA, USA). The GC-MS system was equipped with a HP-5MS capillary column (30.00 m × 0.32 mm × 0.25 µm). Mass spectra were obtained by electron ionization (EI) at 70 eV with a spectra range of 50 to 500 *m*/*z*. The injector and detector temperatures were operated at 150 °C and 250 °C, respectively. The oven temperature was maintained at 80 °C for 4 min, and subsequently raised to 250 °C (5 °C/min) for 10 min. Helium was used as a carrier gas at a flow rate of 1.0 mL/min, at a split ratio of 50:1. The components were identified by comparing Kovats retention indices and their mass spectra with those of the computer mass libraries of The National Institute of Standards and Technology (version 2010, U.S. Department of Commerce, Gaithersburg, MD, USA).

### 3.5. Cell Culture 

HCT116 colon cancer cells and HepG2 liver cancer cells were purchased from Library of Typical Culture of Chinese Academy of Sciences (Shanghai, China). HCT116 cells were cultured in Dulbecco’s modified Eagle’s medium (DMEM; Hyclone, UT, USA) supplemented 10% fetal bovine serum (FBS) and 1% penicillin/streptomycin (Hyclone, UT, USA). HepG2 cells were cultured in MEM containing 10% FBS and 1% penicillin/streptomycin (Hyclone, UT, USA). Above-mentioned cells were maintained in 25 cm^2^ cell culture flasks in a humidified atmosphere containing 5% CO_2_ at 37 °C. Cells were fed until 90% confluence and the confluent cells were washed twice with phosphate buffered saline (PBS), treated with 0.25% trypsin (Invitrogen, MA, USA) for about 1 min, and incubated at 37 °C. When the cells were contracted and rounded under the microscope, FBS (Hyclone, UT, USA) containing medium was added, centrifuged at 200 g for 3 min, and subcultured at a split ratio of 1:3.

### 3.6. Antiproliferative Activity Test of GOEO

Cell proliferation inhibition rate of GOEO was evaluated by Cell Counting Kit-8 (CCK-8) assay [30]. GOEO (50 μL) was added to the medium and mixed well. The mixture was diluted in DMSO to prepare solutions at a concentration of 0.8, 0.7, 0.6, 0.5, 0.4, 0.3, 0.2, 0.1, and 0.0 μL/mL, respectively. The cells were placed into 96-well plates (3 × 10^3^ cells/well). After 24 h, 100 μL of GOEO at different concentrations was added, and incubation continued for 48 h at 37 °C in a CO_2_ incubator. Then, medium in the 96-well plate was disposed of. A 100 μL of CCK-8 test solution (DojinDo, Japan) was added and incubated for 2 h at 37 °C. The optical density (OD) for each well was measured at 450 nm using a microplate reader (BioTek, Winooski, VT, USA). The cell viability rate at different concentrations of GOEO treatment was calculated according to the formula
Viability rate (%) = (OD_sample_ − OD_blank_)/(OD_control_ − OD_blank_) × 100%.(3)

### 3.7. Effect of GOEO on Cancer Cell Migration

Transwell migration assay was performed using transwell chambers [31]. HepG2 and HCT116 cells were transferred to 3–5 passages, treated with 0.3 μL/mL of GOEO for 24 h, and then collected after treatment with trypsin (without EDTA) at room temperature. After centrifugation at 150× *g* for 5 min, the cells were collected, counted, and adjusted to a concentration of 10^5^ cells/mL. Cell suspension (200 μL) was placed in the top chamber of the transwell migration chambers and 600 μL of medium containing 10% FBS was placed in the bottom of the 24-well plate. After incubating for 24 h, cells were fixed by 800 μL of methanol for 30 min and stained with 0.1% crystal violet for 20 min at room temperature. After rinsing three times with PBS, the upper chamber liquid was removed, and the cells on the surface of the upper chamber membrane were carefully wiped off with a cotton swab. Migrated cells on the lower membrane surface photographed and counted. The experiment was repeated at least three times. The migration inhibition rate was calculated according to the formula
Migration inhibition rate (%) = (the number of migration in the control group − the number of migration in the treatment group)/the number of migrations in the control group × 100%.(4)

### 3.8. Statistical Analysis

The mean and standard deviation of three experiments were determined. Statistical analyses of the differences between mean values obtained for experimental groups were calculated using IBM SPSS Statistics 23.0. (IBM Corp. Released 2015. IBM SPSS Statistics for Windows, Version 23.0. Armonk, NY, USA). *p*-values < 0.05 were regarded as significant, *p*-values < 0.01 as very significant, and *p*-values < 0.001 as highly significant.

## 4. Conclusions

In this study, the peel essential oil of ‘Gannanzao’, a new variety from the bud mutation of ‘Newhall’ navel orange, was extracted by hydrodistillation, and the optimum conditions for extraction by response surface design were as follows—liquid material ratio of 8.4:1 (mL/g), NaCl concentration of 5.3%, and distillation time of 3.5 h. Under these conditions, the extraction yield was 2.14%. GC-MS detected thirty-nine chemical components in GOEO. Efficient extraction and identification of EO would be helpful to create new economic value and reduce waste. The in vitro test showed a strong dose-dependent antiproliferative activity of GOEO on HepG2 and HCT116 cancer cells. GOEO at the concentration of 0.3 μL/mL of can inhibit migration of both cancer cells. Further study of the bioactivity of orange EO might be helpful in developing new anticancer agents.

## Figures and Tables

**Figure 1 molecules-24-00499-f001:**
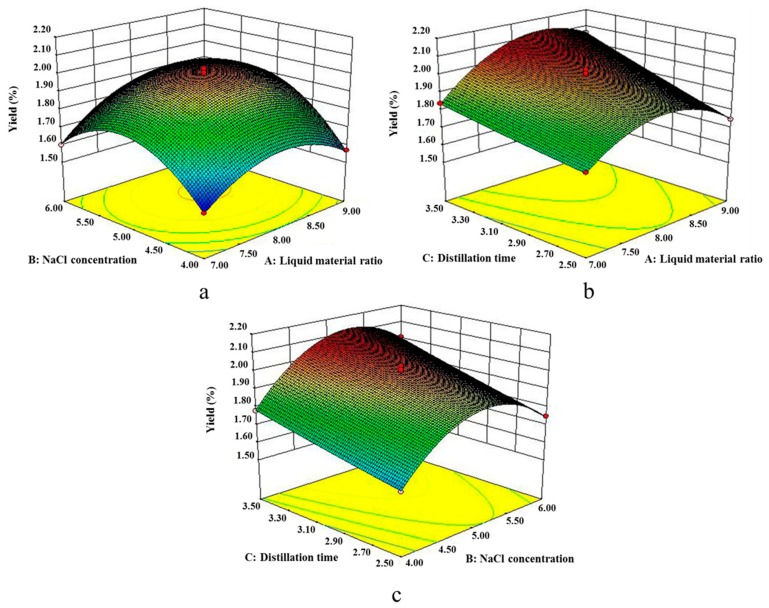
Response surface plot of (**a**) B and A; (**b**) C and A; (**c**) C and B, and their mutual interaction onthe yield of GOEO. A, B, and C represent liquid material ratio, NaCl concentration, and distillation time, respectively.

**Figure 2 molecules-24-00499-f002:**
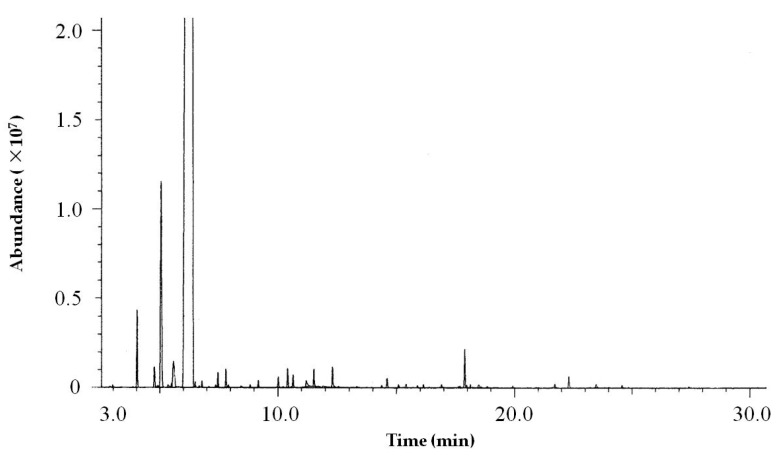
Total ion chromatogram (TIC) of GOEO.

**Figure 3 molecules-24-00499-f003:**
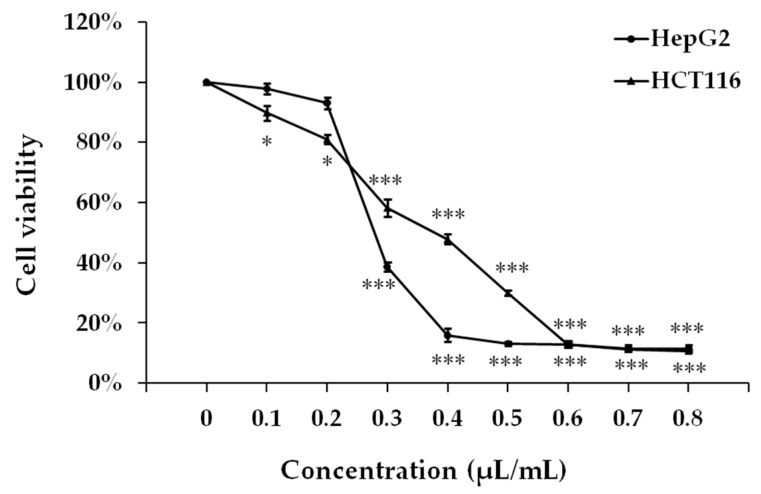
Effect of GOEO on cell viability in cancer cell lines HepG2 and HCT116. The significant differences of cell viability on different concentration compared with control (control group was set to 100%).

**Figure 4 molecules-24-00499-f004:**
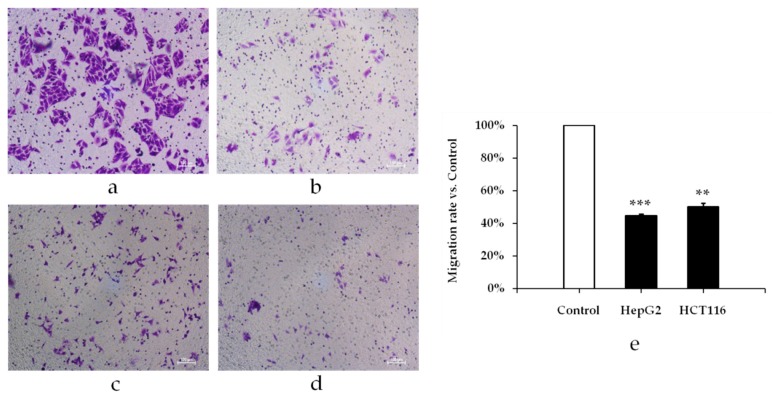
The effect of GOEO on migration of cancer cells. (**a**) HepG2 cells; (**b**) HepG2 cells treated with 0.3 μL/mL of GOEO; (**c**) HCT116 cells; (**d**) HCT116 cells treated with 0.3 μL/mL of GOEO; (**e**) Percentage of migration cells compared with the control group. The cell mobility of the control was set to 100%.

**Table 1 molecules-24-00499-t001:** The Box–Behnken experimental design and data.

Run	A: Liquid Material Ratio (mL/g)	B: NaCl Concentration (%)	C: Distillation Time (h)	Yield (%)
1	−1 (7:1)	0 (5.0)	1 (3.5)	1.84
2	0 (8:1)	0 (5.0)	0 (3.0)	2.01
3	0 (8:1)	1 (6.0)	−1 (2.5)	1.75
4	1 (9:1)	1 (6.0)	0 (3.0)	1.81
5	−1 (7:1)	1 (6.0)	0 (3.0)	1.60
6	0 (8:1)	1 (6.0)	1 (3.5)	2.01
7	0 (8:1)	0 (5.0)	0 (3.0)	1.97
8	0 (8:1)	0 (5.0)	0 (3.0)	2.00
9	0 (8:1)	0 (5.0)	0 (3.0)	2.03
10	1 (9:1)	0 (5.0)	1 (3.5)	2.06
11	−1 (7:1)	0 (5.0)	−1 (2.5)	1.72
12	0 (8:1)	−1 (4.0)	1 (3.5)	1.78
13	0 (8:1)	−1 (4.0)	−1 (2.5)	1.61
14	0 (8:1)	0 (5.0)	0 (3.0)	1.98
15	1 (9:1)	0 (5.0)	−1 (2.5)	1.75
16	1 (9:1)	−1 (4.0)	0 (3.0)	1.57
17	−1 (7:1)	−1 (4.0)	0 (3.0)	1.51

**Table 2 molecules-24-00499-t002:** ANOVA analysis for ‘Gannanzao’ orange essential oil (GOEO) extraction.

Source	Sum of Squares	df	Mean Square	*F*-Value	*p*-Value
Model	0.52	9	0.058	161.29	<0.0001 ***
A	0.034	1	0.034	93.52	<0.0001 ***
B	0.061	1	0.061	169.47	<0.0001 ***
C	0.092	1	0.092	255.79	<0.0001 ***
AB	5.625 × 10^−3^	1	5.625 × 10^−3^	15.56	0.0056 **
AC	9.025 × 10^−3^	1	9.025 × 10^−3^	24.97	0.0016 **
BC	2.025 × 10^−3^	1	2.025 × 10^−3^	5.60	0.0498 *
A^2^	0.11	1	0.11	299.16	<0.0001 ***
B^2^	0.20	1	0.20	539.76	<0.0001 ***
C^2^	9.500 × 10^−5^	1	9.500 × 10^−5^	0.26	0.6239
Residual	2.530 × 10^−3^	7	3.614 × 10^−4^		
Lack of Fit	2.500 × 10^−4^	3	8.333 × 10^−5^	0.15	0.9269
Pure Error	2.280 × 10^−3^	4	5.700 × 10^−4^		
Cor Total	0.53	16			
R^2^ = 0.9952					
Adj R^2^ = 0.9890					
Pred R^2^ = 0.9857					
Adeqprecision = 38.063					

Note: * significant at *p* < 0.05, ** very significant at *p* < 0.01, *** highly significant at *p* < 0.001.

**Table 3 molecules-24-00499-t003:** Chemical composition of GOEO.

No.	RI ^a^	Components	Content (%)
1	937	α-Pinene	1.14
2	975	Sabinene	0.39
3	983	β-Pinene	0.05
4	991	β-Myrcene	4.93
5	1005	Octanal	0.06
6	1011	α-Phellandrene	0.08
7	1014	3-Carene	0.97
8	1047	Limonene	88.07
9	1051	β-Phellandrene	0.06
10	1057	*cis*-β-Ocimene	0.02
11	1062	γ-Terpinene	0.08
12	1089	Terpinolene	0.21
13	1102	Linalool	0.26
14	1105	Nonanal	0.04
15	1125	*trans*-*p*-Mentha-2,8-dien-1-ol	0.02
16	1139	Limonene1,2-epoxide	0.05
17	1152	Citronellal	0.11
18	1183	Terpinen-4-ol	0.16
19	1197	α-Terpineol	0.29
20	1205	Decanal	0.19
21	1226	Nerol	0.18
22	1238	β-Citral	0.27
23	1268	α-Citral	0.36
24	1306	Undecanal	0.02
25	1347	Citronellyl acetate	0.04
26	1356	Neryl acetate	0.15
27	1375	Geranyl acetate	0.07
28	1388	β-Elemene	0.08
29	1407	Dodecanal	0.04
30	1419	Caryophyllene	0.05
31	1450	β-Farnesene	0.06
32	1480	Germacrene D	0.02
33	1491	Valencene	0.60
34	1495	α-Selinene	0.05
35	1501	α-Farnesene	0.05
36	1581	Caryophyllene oxide	0.03
37	1691	β-Sinensal	0.21
38	1761	α-Sinensal	0.07
39	1801	Nootkatone	0.06
Total			99.59

^a^ retention indices determined on HP-5 column, using the homologous series of *n*-alkanes (C8–C20).

**Table 4 molecules-24-00499-t004:** Variables and levels of response surface design.

Factors	Variables	Coded Levels		
		−1	0	1
A	Liquid material ratio (mL/g)	7:1	8:1	9:1
B	NaCl Concentration (%)	4.0	5.0	6.0
C	Distillation time (h)	2.5	3.0	3.5

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
