# Peer review of "Extraction of ‘Gannanzao’ Orange Peel Essential Oil by Response Surface Methodology and its Effect on Cancer Cell Proliferation and Migration"

_molecules, 2019, doi:10.3390/molecules24030499_

Reviewer 1 Report

Introduction line 37: Essential oil is not a secondary metabolite

line 52: change to "carotenoids"

line 54: change to "have become"

lines 75-76: revise p values

Figure 3: is it possible to calculate an IC50 value?

Figures 3 and 4: please add a reference standard to better compare the data

line 183: the discussion about anticancer activity of some EO components has been made by Zengin et al Molecules 2018, 23, 3266; doi:10.3390/molecules23123266

lines 240 and 257: please convert "rpm" in "g"

In the discussion a brief comparison with the previous Newhall navel orange should be done in terms of terpenoid composition

Author Response

Dear editors and reviewers,Thank you very much for your valuable comments on our manuscript. We have revised the manuscript according

your suggestions. We also corrected some errors in the former manuscript. All revisions were clearly highlighted using the "Track Changes"

function in Microsoft Word, so that changes are easily visible to you.

Point 1: Introduction line 37: Essential oil is not a secondary metabolite

Response 1: You are right. Some components of essential oil may not be secondary metabolite, so we deleted ‘secondary metabolite of’.

Point 2: line 52: change to "carotenoids"

Response 2: We have changed according to your suggestion.

Point 3: line 54: change to "have become"

Response 3 : We have changed according to your suggestion.

Point 4: lines 75-76: revise p values

Response 4: From Table 2, we can see p value was less than 0.0001, when the p value was less than 0.001, we regarded as highly significant.

So we used p < 0.0001 < 0.001 to express the result. Because p < 0.001 was enough to show highly significant, so we have revised it according

to your suggestion.

Point 5: Figure 3: is it possible to calculate an IC50 value?

Response 5: We used GraphPad Prism™ (Version 5.00) software (GraphPad Software, San Diego, CA, USA) to calculate IC50 values. IC50 value for HepG2 and HCT116 was 0.29 μL/mL and 0.35 μL/mL, respectively. We have added a sentence in Section 2.3, line 160-162 in the new manuscript.

Point 6: Figures 3 and 4: please add a reference standard to better compare the data

Response 6: We have added one point (EO concentration of 0 μL/mL) in figure 3. The significant differences of cell viability on different concentration compared with control (control group was set to 100%). The OD data of the sample were compared with OD data of the control. We have modified Figure 4 to show the percentage of migration cells compared with the control group. In order to better compare the data, the cell mobility of the control was set to 100%.

Point 7: line 183: the discussion about anticancer activity of some EO components has been made by Zengin et al Molecules 2018, 23, 3266; doi:10.3390/molecules23123266

Response 7: We have added this sentence to the new manuscript (line 195-196) and cited it as reference 24.

Point 8: lines 240 and 257: please convert "rpm" in "g"

Response 8: We have changed according to your suggestion (lines 253 and 270 in the new manuscript).

Point 9: In the discussion a brief comparison with the previous Newhall navel orange should be done in terms of terpenoid composition

Response 9: We have added a brief comparison with the previous Newhall navel orange (lines 139-145 in the new manuscript):

Compared with ‘Newhall’ orange EO (NOEO) prepared by hydro-distillation, the total amount of monoterpenes in GOEO was 95.94%, higher than

NOEO (89.87%). Limonene content of GOEO was about 6.4% higher than NOEO (81.62%). Percentage of oxygenated monoterpenes in

GOEO (1.96%) was much lower than NOEO (6.94%). linalool (0.26%) content in GOEO was much lower than NOEO (2.03%). Sesquiterpenes

and oxygenated sesquiterpenes presented very low content in both EOs (0.81% and 0.17% for NOEO, 0.91% and 0.37% for GOEO, respectively).

Other revisions:

1.     I have changed email address according to your suggestion.

2.     The unit of concentration of EO should be μL/mL, not μg/mL, very sorry for this mistake.

3.     We have corrected the small error of viability rate. in line 21 and 158, 13.8% was changed to 13.2%, in line 160, 12.2% was

changed 13.0%.

Reviewer 2 Report

This is an interesting manuscript that describes optimising the extraction of GOEO and cell migrating activity. What is missing for me is comparison

data to Newhall variety, which the researchers had previously worked on (ref 18), and statistically rigour in Table 1 results.

Specific questions and suggestions:

Line 66: Suggest '...literature reports, we..."

Line 71. Table 1: From line 268 it states that experiments were done in triplicate. Is this correct for Table 1?

What is the error and how significant are the yield differences taking this error into account?

Were experiments tried on different days? Triplicate analyses on three different days would give more confidence.

What is the statistical differences (oil and components) between individual oranges? Do these affect the conclusions?

Line 167: How does the bioactivity compare to Newhall oranges (Fig 4e)?

Line 175: "... and colon cancer."

Line 282: I seriously doubt that oranges will be the source of a new anticancer agent. The bioactivity of the oil and components has been

well studied.

Author Response

Dear editors and reviewers,Thank you very much for your valuable comments on our manuscript. We have revised the manuscript according

your suggestions. We also corrected some errors in the former manuscript. All revisions were clearly highlighted using the "Track Changes"

function in Microsoft Word, so that changes are easily visible to you.

Point 1: Line 66: Suggest '...literature reports, we..."

Response 1: We have corrected it according your suggestion.

Point 2: Line 71. Table 1: From line 268 it states that experiments were done in triplicate. Is this correct for Table 1?

Response 2 : Yes.

Point 3: What is the error and how significant are the yield differences taking this error into account?

Response 3: The relative standard deviation (RSD%) of the three replicate experiments was less than 0.15%. We use the mean value and did

not take the small error into account.

Point 4: Were experiments tried on different days? Triplicate analyses on three different days would give more confidence.

Response 4: Most experiments were tried on different days, sometimes we tried one in the morning, another in the afternoon or evening and

the third on the next day.

Point 5: What is the statistical differences (oil and components) between individual oranges? Do these affect the conclusions?

Response 5: The composition of orange essential oil (EO) varies markedly according to variety, seasonality, geographic origin, ripeness of the fruit, extraction method and also an interaction of various factors. There may be large differences (oil and components) between individual oranges.

The material we used to extract EO was the oranges from the same location that were mature and available for extracting juice or fresh

consumption. We used GC-MS to detect the relative contents of the oil components. The major component such as limonene, its content did

not have large differences in individual oranges; however, some minor components in individual oranges may have large differences. We collect oranges in large quantities and mix them together and use the same method to extract the essential oil, the components of the sample was

relatively stable. However, individual oranges may present big differences, which may affect the conclusions.

Point 6: Line 167: How does the bioactivity compare to Newhall oranges (Fig 4e)?

Response 6: As we indicated in the manuscript, ‘Gannanzao’ is a new variety selected from the bud mutation of ‘Newhall’ navel orange, which

has the characteristics of early maturity (the harvest time is one month earlier than the latter), excellent flavor, and easy peeling’. Comparing their bioactivity will be very interesting. Unfortunately, we only test the bioactivity ‘Gannanzao’. The reasons are the following:

1.     The output of ‘Newhall’ navel orange was over one million tons, it was easy to use cold pressing method to extract EO in large industrial scale. We have published a paper about anticancer activity of ‘Newhall’ EO prepared by molecular distillation of the cold pressed EO.

2.     ‘Gannanzao’ is a new variety and the output was much lower than ‘Newhall’ orange. It was not convenient to extract EO by cold pressing method. That’s why in this paper, we focus on using the hydro-distillation method to extract EO and optimising the conditions. It took us more

time and energy to optimize extraction of ‘Gannanzao’ EO but less time to do bioactivity test, only preliminary result of bioactivity was obtained. Because ‘Newhall’ EO and ‘Gannanzao’ EO were made by totally different method, their components were quite different. For example, perillyl alcohol, an important anticancer agent, was found in cold pressed ‘Newhall’ orange EO, but not found in hydro-distilled ‘Gannanzao’ EO. We thought it is not reasonable to compare bioactivity of EOs from both different variety and different extraction method, that’s why we did not do

the comparison of their bioactivity.

3.     Unfortunately, we cannot prepare EO of ‘Newhall’ by hydro-distillation method now because ‘Newhall’ oranges were harvested before December in 2018 and no fresh ‘Newhall’ oranges are available.

4.     We do have a plan to compare the bioactivity of this two oranges in the future, for example, prepare EO samples by the same method, use them to treat the same cell line to see what happened, and test bioactivity of their important components. This will involve very careful plan and

a lot of work.

It is our fault that we did not realize that we should do bioactivity comparison in time. However, we can do a much thorough work in the future.

Thank you for the nice idea and suggestion.

Point 7: Line 175: "... and colon cancer."

Response 7: We have corrected it according your suggestion.

Point 8: Line 282: I seriously doubt that oranges will be the source of a new anticancer agent. The bioactivity of the oil and components has been

well studied.

Response 8: Because the bioactivity test was preliminary in this paper, it is too early to say that orange EO will be the source of a new anticancer agent. So we change the sentence as ‘Further study of the bioactivity of orange EO might be help to develop new anticancer agents’ (line 289 in

the new manuscript)

The bioactivity of citrus oil and components has been well studied. There have been much clinical advances in anticancer essential oils. Components as limonene, elemene, and perillyl alcohol have been used to do phase I and phase II study, and some promising results have been achieved. So it is very interesting to continue exploring anticancer activity of orange EO and its components.

Other revisions:

1.     I have changed email address according to your suggestion.

2.     The concentration of EO should be μL/mL, not μg/mL, very sorry for this big mistake.

3.     We have corrected the small error of viability rate. in line 21 and 158, 13.8% was changed to 13.2%, in line 160, 12.2% was changed 13.0%.

4.     We have revised fig 3 and fig 4 according to another reviewer’s suggestion.

Round  2

Reviewer 1 Report

the manuscript is now suitable for publication 

Reviewer 2 Report

I am happy with the author's responses and would recommend publication.